Untangling the taxonomic knot of Croton anomalus (Euphorbiaceae), a Neotropical dry forest shrub

Rossine Yuri yuri.rossine@ufpe.br yurirossine@gmail.com 1
Riina Ricarda rriina@rjb.csic.es 2
Silva Otávio L.M. 3 4
Louzada Rafael 1
1 Departamento de Botânica, Universidade Federal de Pernambuco , Recife , Pernambuco , Brazil
2 Real Jardín Botánico (RJB), CSIC , Madrid , Spain
3 Departamento de Botânica, Instituto de Biociências, Universidade de São Paulo , São Paulo , Brazil
4 Herbarium SP, Instituto de Pesquisa Ambientais , São Paulo , Brazil
Casazza Gabriele
Electronic publication date: 2025 Apr 1
Publication date: 2025
Volume: 13
Electronic Location ID: e19176
Received 2024 Oct 9; Accepted 2025 Feb 24
Copyright: ©2025 Rossine et al.
Copyright year: 2025
Copyright holder: Rossine et al.
License: This is an open access article distributed under the terms of the Creative Commons Attribution License, which permits unrestricted use, distribution, reproduction and adaptation in any medium and for any purpose provided that it is properly attributed. For attribution, the original author(s), title, publication source (PeerJ) and either DOI or URL of the article must be cited.
License URL: https://creativecommons.org/licenses/by/4.0/

Keywords: Croton section Lasiogyne, Dry forest, Integrative taxonomy, Marmeleiros, Neotropics

Funding: Coordenação de Aperfeiçoamento de Pessoal de Nível Superior—CAPES PROEX 88887.637810/2021-00 PrInt 88887.837805/2023-00 International Association for Plant Taxonomy 2021/08545-2 2022/12597-0 Universidade de São Paulo, Programa de Apoio a Novos Docentes 22.1.09345.01.2 MCIN/AEI/10.13039/501100011033 PID2019-108109GB-I00 FEDER The Coordenação de Aperfeiçoamento de Pessoal de Nível Superior—CAPES (processes PROEX 88887.637810/2021-00 and PrInt 88887.837805/2023-00), and International Association for Plant Taxonomy (IAPT) granted YR. Fundação de Apoio à Pesquisa do Estado de São Paulo—FAPESP (process numbers 2021/08545-2 and 2022/12597-0), and Universidade de São Paulo, Programa de Apoio a Novos Docentes (process 22.1.09345.01.2) granted to OLMS. RR was supported by grant PID2019-108109GB-I00 from MCIN/AEI/10.13039/501100011033/ and FEDER “A way to make Europe”. The funders had no role in study design, data collection and analysis, decision to publish, or preparation of the manuscript.

==============================
Croton anomalus was described by Henri Pittier in 1930 from a collection made in Estado Lara, Venezuela, and the use of the name has so far been restricted to several states in this country. A reevaluation of the species has led to its recircumscription and recognition in several other countries, including Bolivia, Brazil, Colombia, French Guiana, Guyana, Mexico, and Suriname. Previously, it was confused with species referred to here as the “Croton anomalus group”, namely C. acapulcensis, C. blanchetianus, C. chiapensis, C. jacobinensis (=C. sonderianus), and C. stahelianus. We integrated morphological, phylogenetic, and ecological evidence to understand species limits and relationships within the Croton anomalus group. We first studied ca. 650 herbarium specimens covering the geographic range of the group, and we inferred species phylogenetic relationships using DNA sequences from the nuclear and plastid regions (ITS and trnL-F). We also used ecological niche modeling to infer potential suitable areas for the occurrence of the studied species and to determine the variables that most contribute to their distribution model. Both morphological and phylogenetic data provide evidence for the synonymization of C. acapulcensis, C. chiapensis, and C. stahelianus under C. anomalus. On the other hand, our results support the recognition of C. blanchetianus and C. jacobinensis as two independent lineages, both distinct from C. anomalus. An emended description of C. anomalus is provided, as well as the designation of lectotypes, illustrations, updates of distribution data, and morphological comparisons with closely related species. Regarding niche modeling, annual precipitation and the precipitation of the warmest quarter were the most important variables explaining species distributions. Croton anomalus showed suitable areas in most seasonally dry tropical forests in the Neotropics, while C. blanchetianus and C. jacobinensis had their most suitable areas restricted to the Caatinga Dry Forest (Brazil), and Caatinga + northern South America, respectively. Our study shows the importance of taxonomic revisions using integrative approaches to disentangling species boundaries and to elucidate their biogeography and conservation status.

Introduction

The delimitation of morphologically very similar species is a long-standing challenge in taxonomy and it is a common problem in megadiverse lineages. Properly recognizing such species is crucial for understanding their distributions, assessing conservation status, and more accurately estimating biodiversity (Bickford et al., 2007; Padial et al., 2010). Since occurrence data are one of the most important pieces of evidence when designating new species and identifying those already described, delimitation is more difficult when the putative taxa present disjunct distributions (Padial et al., 2010; Medina et al., 2012; Salvador-Montoya et al., 2015; Guimarães et al., 2021; Sassone et al., 2021), which in some cases may be artifactual due to poorly explored areas or collection gaps (Feeley, 2015; Riina et al., 2018; Ondo et al., 2024). Thus, the use of an integrative approach becomes a better alternative to achieve more accurate taxonomic decisions (Padial et al., 2010; Frajman et al., 2019).

Croton L. has been identified to be one of the most species-rich genera in many areas of seasonally dry tropical forests (SDTF) (Carneiro-Torres, 2009; DRYFLOR et al., 2016; Marcelo-Peña et al., 2016; Quintana et al., 2017). The genus is particularly important in the Caatinga SDTF, where 70 species have been recorded so far (Caruzo et al., 2020). Although the SDTF biome has long been underappreciated by the scientific community when compared to savannas and tropical rainforests, which have received more research attention (Moonlight et al., 2020), recent studies have increased our floristic knowledge of this important biome (DRYFLOR et al., 2016; Escribano-Avila et al., 2017; Quintana et al., 2017; BFG-The Brazil Flora Group et al., 2021; Carrión et al., 2022; Wurdack, 2023). However, the diversity of the Caatinga, the largest area of SDTF, remains understudied (Queiroz et al., 2017; Fernandes, Cardoso & Queiroz, 2020), suggesting that the number of Croton species in the Caatinga is likely underestimated.

The taxonomic knowledge of Croton, a genus with over 1,200 species (Moonlight et al., 2024), is still incomplete and problematic for many species groups within it (Berry et al., 2005; Van Ee, Riina & Berry, 2011). For example, C. sonderianus Müll.Arg. is a frequent name in studies spanning taxonomy, phytochemistry, pharmacology, and ecology in Brazil (e.g., Craveiro et al., 1981; Craveiro & Silveira, 1982; de Araújo Lucena & de Sales, 2006; Lôbo, Tabarelli & Leal, 2011; de Souza et al., 2017). However, the application of this name in northeastern Brazil to various Croton species, particularly those locally known as ‘marmeleiros’ (Carneiro-Torres, 2009), poses a taxonomic challenge. Despite the proposal of Gomes, Sales & Melo (2010) that C. sonderianus should be synonymized under C. jacobinensis Baill., the use of the former persists in the scientific literature, including Institute of Research Rio de Janeiro Botanical Garden (2020). In addition, in numerous cases the name C. sonderianus has also been misapplied to specimens of C. blanchetianus Baill.

After an extensive revision of specimens identified as C. sonderianus, we realized that a third species (C. anomalus Pittier) was involved, which had never been cited for Brazil. Croton anomalus was published nearly a century ago (Pittier, 1930), and it was known only from Venezuela, where it has been reported for several states (Carabobo, Distrito Federal, Lara, Nueva Esparta, Yaracuy y Zulia) by Hokche, Berry & Huber (2008). Additionally, three other species, very similar to C. anomalus, were noted during our ongoing revision of C. sect. Lasiogyne (Klotzsch) Baill. and were also considered for this study. These species are C. acapulcensis Mart.Gord. & J. Jiménez Ram. and C. chiapensis Lundell, both from Mexico, and C. stahelianus Lanj., known from Suriname (Lanjouw, 1931; Lundell, 1942; Gordillo & Ramírez, 1990; Funk et al., 2007). Based on these morphological observations, we proposed the following hypotheses: (i) Croton blanchetianus and C. jacobinensis are separate lineages independent of C. anomalus; and (ii) C. anomalus includes C. acapulcensis, C. chiapensis, and C. stahelianus as synonyms, and has a broad geographic range.

The taxonomic riddle concerning C. anomalus and other names/species in the Croton anomalus group is discussed and disentangled here. We use morphological, ecological, geographic, and phylogenetic data to elucidate the taxonomy of this group. We also employ ecological niche modeling to uncover potential suitable areas for the occurrence of the studied species. An amended description and designation of a lectotype for C. anomalus, distribution maps, illustrations, preliminary conservation assessments, and a historical discussion about this taxonomic complex are provided.

Materials and Methods

Sampling

We analyzed ca. 650 herbarium specimens identified as Croton anomalus, C. acapulcensis, C. chiapensis, C. blanchetianus (see Appendix 1 in Rossine et al., 2023), C. jacobinensis (see Appendix 1 in Rossine et al., 2023), C. sonderianus, and C. stahelianus. The specimens deposited in A, CA, CEPEC, EAC, HESBRA (not indexed), HUEFS, HUESB, HST, IPA, K, MA, MICH, MO, MOSS, NY, PEUFR, UFP, US, VEN, and W were analyzed, and complemented by virtual specimens from other herbaria (CAS, DAV, F, MAC, MEXU, P, SP, SPF, TEX, U, and USZ) available at SpeciesLink (https://specieslink.net/search/), Southeast Regional Network of Expertise and Collections—SERNEC (https://sernecportal.org/portal/) or directly from herbarium online data portals. Verifiable field records of Croton anomalus from iNaturalist, identified by Croton taxonomists, were included when their localities were not represented by herbarium records. All herbarium acronyms follow Thiers (2024).

For the phylogenetic analyses, we included a total 37 specimens, representing 29 putative species (28 of Croton, and Brasiliocroton muricatus Riina & Cordeiro as the outgroup), of which 16 samples were part of the Croton anomalus group (Table 1). Unfortunately, we were not able to sample C. chiapensis for phylogenetic analyses due to the lack of material for DNA extraction. The species is known only from the type specimens collected over 85 years ago, and a from second gathering collected over 40 years ago.

Table 1 Metadata of the specimens used for phylogenetic reconstruction, including Genbank accession numbers.

Species	Locality	Voucher	Accession number	
			ITS	trnL-F	
Brasiliocroton muricatus	Bahia, Brazil	Carneiro-Torres 1000	KF208629	KF208632	
Croton acapulcensis*	Jalisco, Mexico	Gentry 74399	EU477862	EU478122	
Croton anomalus*	Sucre, Venezuela	Riina 1843	PQ350252	PQ458518	
Croton anomalus*	Santa Cruz, Bolivia	Vargas 1915	PQ350253	PQ458519	
Croton anomalus*	Bahia, Brazil	Couto 58	PQ350254	PQ458520	
Croton anomalus*	Ceará, Brazil	Nunes s.n. (HUEFS 111097)	PQ350256	PQ458522	
Croton anomalus*	Piauí, Brazil	Melo 8775	PQ350257	PQ458523	
Croton anomalus*	Bahia, Brazil	Souza-Silva 283	PQ350255	PQ458521	
Croton anomalus*	Espírito Santo, Brazil	Marinero 1164	PQ350258	PQ458524	
Croton argenteus	Guanacaste, Costa Rica	van Ee 297	EU478094	EU497702	
Croton argentinus	Cordoba, Argentina	van Ee 644	HM071943	HM071965	
Croton argyrophyllus	Pernambuco, Brazil	van Ee 476	HM564076	HM564213	
Croton astroites	Puerto Rico	van Ee 537	EU586901	EU586955	
Croton blanchetianus*	Bahia, Brazil	Queiroz 8057	PQ350259	PQ458525	
Croton blanchetianus*	Bahia, Brazil	Rodarte 124	PQ350260	PQ458526	
Croton blanchetianus*	Pernambuco, Brazil	Rossine 234	PQ350261	PQ458527	
Croton blanchetianus*	Rio Grande do Norte, Brazil	Rossine 133	PQ350262	PQ458528	
Croton campanulatus	Rio de Janeiro, Brazil	Caruzo 93	MW263138	MW266678	
Croton elliottii	Florida, USA	Wurdack 455	AY971208	AY971297	
Croton floribundus	Pernambuco, Brazil	Rossine 422	PQ350263	PQ458529	
Croton fuscescens	São Paulo, Brazil	van Ee 502	HM564081	HM564217	
Croton glandulosus	Puerto Rico	van Ee 550	EU478068	EU478159	
Croton heptalon	Veracruz, Mexico	Breckon 2013	FJ614722	FJ614783	
Croton jacobinensis*	Bahia, Brazil	Carneiro-Torres 789	HM044795	HM044776	
Croton jacobinensis*	Pernambuco, Brazil	Rossine 248	PQ350264	PQ458530	
Croton jacobinensis*	Ceará, Brazil	Rossine 600	PQ350265	PQ458531	
Croton leucophyllus	Tamaulipas, Mexico	Breckon 2002	EU478108	FJ614786	
Croton luetzelburgii	Bahia, Brazil	Conceição 1457	HM564087	HM564222	
Croton matourensis	Amazonas, Brazil	van Ee 492	EU478096	EU497720	
Croton pachypodus	Altos de Campana, Panama	Montenegro 1111	EF421789	EF408131	
Croton pedicellatus	Cundinamarca, Colombia	Plowman 3766	FJ614766	FJ614804	
Croton sacaquinha	Pará, Brazil	Caruzo 97	HM044802	HM044782	
Croton sampatik	Pasco, Peru	Riina 1447	EF421792	EF408133	
Croton stahelianus*	Suriname	Mennega 62	PQ350266	PQ458532	
Croton tricolor	São Paulo, Brazil	Caruzo 87	EF421752	EF408125	
Croton watsonii	Veracruz, Mexico	van Ee 527	EU477882	EU478125	
Croton yucatanensis	Yucatán, Mexico	van Ee 121	DQ227537	DQ227569	
Notes.

Names in bold are members of Croton section Lasiogyne, and those marked with an asterisk (*) are part of the Croton anomalus group. Newly generated sequences are those with accession numbers starting with PQ.

DNA extraction, amplification, sequencing, and alignment

Total genomic DNA was extracted according to the CTAB protocol (Doyle, 1991). The nuclear ribosomal internal transcribed spacer (ITS) region, and the plastid intergenic spacer trnL-trnF (hereafter called trnL-F) were amplified using the PCR settings of Masa-Iranzo et al. (2021) and sent to Macrogen (Macrogen, Madrid) for sequencing. Primers and references are given in Table 2. The aforementioned regions have been widely used and proven to be informative in previous studies of Croton (e.g., Van Ee, Riina & Berry, 2011; Masa-Iranzo et al., 2021; Riina et al., 2021). Sequences were assembled and aligned with MAFFT (default parameters), followed by manual editing in Unipro UGENE (Okonechnikov et al., 2012). Summary statistics of each data matrix were estimated in PAUP v.4.0a169 (Swofford, 2003).

Table 2 Primers used in the present study followed by their sequences (5′–3′) and references.

Primer name	Primer sequence (5′–3′)	Reference	
ITSw1	CCTTATCATTTAGAGGAAGGAG	Silva, Riina & Cordeiro (2020)	
ITSp2	GCCRAGATATCCGTTGCCGAG	Cheng et al. (2016)	
ITSp3	YGACTCTCGGCAACGGATA	Cheng et al. (2016)	
ITSw2	TATGCTTAAAYTCAGCGGGT	Silva, Riina & Cordeiro (2020)	
trnL-Fc	CGAAATCGGTAGACGCTACG	Taberlet et al. (1991)	
trnL-Fd	GGGGATAGAGGGACTTGAAC	Taberlet et al. (1991)	
trnL-Fe	GGTTCAAGTCCCTCTATCCC	Taberlet et al. (1991)	
trnL-Ff	ATI’TGAACTGGTGACACGAG	Taberlet et al. (1991)	

Phylogenetic analyses

Bayesian Inference (BI) was used to reconstruct phylogenetic relationships of the nuclear and plastid regions individually. We used the threshold of clade support (>0.95 posterior probability (PP) proposed by Alfaro, Zoller & Lutzoni (2003) to account for incongruent clades between topologies. Since there were no highly supported (>0.95 PP) incongruences between the two topologies, a concatenated dataset was generated using SequenceMatrix (Vaidya et al., 2011). The BI was performed in MrBayes v.3.2.7a (Ronquist et al., 2012). The best substitution models were determined in JModelTest2 v.2.1.10 using the Akaike Information Criterion (AIC) and were SYM+I+G for ITS, TIM1+G for trnL-F, and GTR+I+G for the concatenated matrix. These were replaced by the most similar and compatible models implemented in MrBayes: GTR+I+G for ITS, GTR+G for trnL-F, and GTR+I+G for the concatenated matrix. The aforementioned analyses were all run on the XSEDE-CIPRES platform (Miller, Pfeiffer & Schwartz, 2010). Bayesian PP were generated from two runs each of four chains. The parameters set were the following: 30 million generations with sampling in every 1,000th generation; default priors, and discarding the first 25% sampled trees and parameters for burn-in. Estimated sample size (ESS) values (>200) were checked in Tracer v.1.6 (Rambaut & Drummond, 2007).

The phylogenetic trees were visualized and edited in FigTree (Rambaut, 2010) followed by export of a vector edition for publishing. The support values for Bayesian posterior probability (PP) were considered moderate when ≥0.75 to <0.95 PP and strong when ≥0.95 PP.

Taxonomic treatment

Protologues, type specimens, historical and general collections were examined and compared for all names/species treated here. The nomenclatural decisions follow the rules and recommendations of the International Code of Nomenclature for algae, fungi, and plants (Turland et al., 2018). The general terminology used in the taxon descriptions follows Simpson (2006). Specific terminology for section Lasiogyne follows previous works on the group (Rossine et al., 2023) with additions from Pinto-Silva et al. (2023) for trichomes.

Distribution maps, conservation status, and ecological niche modeling

Geographic coordinates were obtained from the labels of herbarium specimens. When coordinates were not available, but the locality was described, the approximate coordinates (included in square brackets) were estimated using Google Earth (https://earth.google.com/web/). Distribution maps were produced in the Quantum Geographic Information System (QGIS) version 3.4.13 (QGIS Development Team, 2020), using cartographic data from OpenDataSoft (https://public.opendatasoft.com) for World administrative boundaries, and the DRYFLOR network (http://www.dryflor.info/) for SDTFs areas. Ecoregion’s descriptions follow Olson et al. (2001). The preliminary assessments of the species conservation status were based on the guidelines from the IUCN Red List Categories and Criteria v.15.1 (IUCN International Union for Conservation of Nature, 2022) under B criteria, applying the estimated values of extent of occurrence (EOO) and area of occupancy (AOO), obtained from GeoCAT (http://geocat.kew.org/), following Bachman et al. (2011).

Using the same set of georeferenced records cited above, we modeled the ecological niche of each putative species using MaxEnt v.3.4.3 (Phillips et al., 2017). Default parameters were used with 10,000 background points (pseudoabsence). Replicates were performed 3 times. The variables used (Table 3) had 30 arc-sec resolution and were: bioclimatic from WorldClim (Fick & Hijmans, 2017), elevation from the Shuttle Radar Topographic Mission (NASA–SRTM), and soil quality (Fischer et al., 2008). The area under the curve (AUC) was used to evaluate the projections.

Table 3 Description of the bioclimatic variables used in the niche modeling analysis of the Croton anomalus group.

Variable name	Description	
alt	Altitude	
bio1	Annual Mean Temperature	
bio2	Mean Diurnal Range (Mean of monthly [max temp - min temp])	
bio3	Isothermality (BIO2/BIO7) (* 100)	
bio4	Temperature Seasonality (standard deviation *100)	
bio5	Max Temperature of Warmest Month	
bio6	Min Temperature of Coldest Month	
bio7	Temperature Annual Range (BIO5-BIO6)	
bio8	Mean Temperature of Wettest Quarter	
bio9	Mean Temperature of Driest Quarter	
bio10	Mean Temperature of Warmest Quarter	
bio11	Mean Temperature of Coldest Quarter	
bio12	Annual Precipitation	
bio13	Precipitation of Wettest Month	
bio14	Precipitation of Driest Month	
bio15	Precipitation Seasonality (Coefficient of Variation)	
bio16	Precipitation of Wettest Quarter	
bio17	Precipitation of Driest Quarter	
bio18	Precipitation of Warmest Quarter	
bio19	Precipitation of Coldest Quarter	
sq1	Nutrient availability	
sq2	Nutrient retention capacity	
sq3	Rooting conditions	
sq4	Oxygen availability to roots	
sq5	Excess salts	
sq6	Toxicity	
sq7	Workability (constraining field management)	

Results

Phylogenetic relationships within the Croton anomalus group

The data matrix included 74 sequences, of which 30 were newly generated in this study (15 of ITS and 15 of trnL-F). Voucher information and GenBank numbers for all the sequences used in the analyses are included in Table 1. The data alignment needed few manual adjustments after the automatic alignment. The aligned matrices of ITS and trnL-F are provided in Files S1 and S2. Summary statistics for each data matrix are shown in Table 4. The ITS region had more variable characters and parsimony informative sites than trnL-F (Table 4). Both nuclear ITS and plastid trnL-F analyses recovered similar topologies on the resulting trees, and no incongruences were found between the two datasets regarding the lineages formed by C. anomalus, C. blanchetianus, and C. jacobinensis (phylogenetic trees not shown, see Figs. S1, S2).

Table 4 Summary statistics of the molecular dataset estimated in PAUP for the Croton anomalus group.

The substitution models were implemented in the analyses using MrBayes.

Dataset	NA	AL	CC	VC	PIC (%)	SM	
ITS	37	625	363	262	159 (25.44)	GTR + I + G	
trnL-F	37	991	839	152	57 (5.75)	GTR + G	
Combined	74	1,616	1,202	414	213 (13.19)	GTR + I + G	
Notes.

NA Number of accessions

AL Aligned length

CC Constant characters

VC Variable characters

PIC Parsimony Informative Characters

SM Substitution model (see ‘Materials and Methods’)

The phylogenetic reconstruction of the concatenated dataset recovers section Lasiogyne as a monophyletic lineage, with strong support (PP = 1; Fig. 1). Similarly, the topology of the backbone of the genus, including 10 other sections (Fig. 1), is in agreement with previous phylogenetic analysis at the genus level (Berry et al., 2005; Van Ee, Riina & Berry, 2011). Section Julocroton was recovered as the sister clade of section Lasiogyne, and section Heptallon as sister of the former two with high support (PP = 1; Fig. 1).

Figure 1 Phylogenetic reconstruction of the Croton anomalus group.

Majority consensus tree obtained from the Bayesian analysis of combined ITS and trnL-F datasets. Names in bold are those newly generated in this study. Accession names in bold correspond to sequences newly generated in this study. For each accession, voucher information (collector and collection number) is given next to the species name.

The concatenated (ITS + trnLF) phylogenetic analysis recovered the specimens of C. blanchetianus and C. jacobinensis as separate monophyletic lineages with high support (PP = 1), both independent of C. anomalus (Fig. 1). Croton acapulcensis and C. stahelianus were recovered in a highly supported clade along with all the C. anomalus specimens (1 PP; Fig. 1).

Taxonomic treatment

Croton anomalus Pittier, J. Wash. Acad. Sci. 20: 4. 1930. –TYPE: VENEZUELA. Lara: Los Rastrojos, between Sarare and Barquisimeto, in bushes, (10°04′29″N, 69°15′31″W), 9.IV.1925, H. Pittier 11757 (lectotype, designated here: A barcode A00047223!; isolectotypes: US barcode US00109497!, VEN barcode VEN6465!). Figs. 2–3.

Figure 2 Main morphological features distinguishing Croton anomalus.

(A) Flowered branch, showing the monopodial branching, disposition of stipules and leaves, and terminal inflorescences. (B–E) Leaf blades, (B) leaf blade with entire margins; (C) stellate-porrect trichomes found on petioles; (D) leaf blade with slightly serrate margins, showing the detail of a section of the margin; (E) leaf blade with irregularly erose margins; (F) narrowly lanceolate stipules; (G) linear (right) and 3-lobed (left) bracts; (H) staminate flower; (I–J) pistillate flower, (I) pistillate flower with 6 sepals; (J) pistillate flower with five sepals, showing the detail of the trichomes on the stigmatic tips; (K) capsule, showing the detail of the inner surface of the sepals with stellate and stellate-porrect trichomes; (L) ventral (left) and dorsal (right) view of a seed.

Figure 3 Field and herbarium images of Croton anomalus.

(A) Shrubby habit (B) general aspect of leaves and inflorescences, (C) stipules, (D) adaxial and (E) abaxial view of the leaf blade; (F) early anthesis of the inflorescences with only pistillate flowers open; (G) branches with unissexual staminate inflorescences; (H) bisexual inflorescence; (I) infructescence and (J) capsule showing the stellate porrect trichomes, and (K) ventral and dorsal view of the seed. Photographs: A, B, F, G (C. Domínguez-Rodríguez); C, D (J. Mota); H (E. S. Velásquez Arellanes), reproduced with permission.

=Croton acapulcensis Mart. Gord. & J. Jiménez Ram., Anales Inst. Biol. Univ. Nac. Autón. México, Bot. 60: 40. 1990. –TYPE: MEXICO. Guerrero: Acapulco, Parque Nacional “El Veladero”, El Mirador, 16°45′00″N, 99° 43′55″W, alt. 350 m, 6.VII.1985, N. Noriega Acosta 599 (holotype: FCME [not seen]; isotype: MEXU barcode MEXU00511553!), syn. nov.

= Croton chiapensis Lundell, Contr. Univ. Michigan Herb. 7: 18. 1942. TYPE: MEXICO. Chiapas: Escuintla, 160 m., VII.1938, E. Matuda 2614 (holotype: MICH barcode MICH1191498!; isotypes: LL barcodes LL00371633! and LL00208230!, F barcode F0056020F!, A barcode A00047090!, MEXU barcode MEXU00078286!, EAP barcode EAP112178!) syn nov.

= Croton stahelianus Lanj., Euphorb. Surinam, 17. 1931. –Type: SURINAME. Upper Koetarie R., 16.X.1926, B.W. n. 7002 = Stahel n. 611 (lectotype, designated here: U barcode U0178808!; isolectotypes: GH barcode GH00047432, IAN barcode IAN049296!, K barcodes K000254419! and K000254420!, RB barcode RB00538444!, U barcode U1617257, US barcode US00109761!) syn nov.

Monoecious shrub, 0.7–2 m, latex not reported, monopodial ramification, branches brown, glabrescent, young ones greenish to yellowish with stellate trichomes. Leaves alternate; stipules linear to narrowly lanceolate, 5–8.8 × 1–1.4 mm, persistent, trichomes stellate; petiole 0.5–4 cm, trichomes stellate to multiradiate; leaf blade oval to elliptic, membranous, 2.5–12 × 1–6 cm, base rounded to cordate, margins entire, irregularly sinuate to slightly serrated, apex acuminate, discolorous, adaxial surface dark green with stellate-porrect trichomes, abaxial surface light green to yellowish with stellate trichomes, venation eucamptodromous to brochidodromous, 4–8 secondary veins, curved, ascendant. Inflorescences terminal or axillary, unisexual staminate or bisexual, flowers congested, 1–5 cm long, trichomes stellate to multiradiate, 3–7 pistillate and 5–13 staminate flowers per inflorescence; bracts linear or 3-lobed, 3–6 mm long, margins entire, apex acuminate, trichomes stellate-porrect, 1 bract per cymule, persistent. Staminate flowers with pedicel 3–4 mm, sepals yellowish, oval, 2 × one mm, margins entire, apex acute, united only near the base, trichomes stellate, indument dense externally and absent to sparse internally; petals white to yellowish, oblong, 3 × one mm, margins entire, apex rounded, trichomes simple; stamens (11-)14–17(-18), filaments ca. two mm, anthers elliptic, 1 × 0.5 mm; nectary disk rounded, unlobed, trichomes simple. Pistillate flowers with pedicels 2–5 mm (up to 10 mm when in fruit), sepals 5–6 (7), equal to slightly unequal in size (when having 7 sepals), light green to whitish externally, dark green internally, oval to widely oval, 2.5–5 × 1.5–2.9, margins entire, apex acute, reduplicate lateral and vertically, 1/5 united at the base, trichomes stellate and stellate-porrect, indument dense externally and sparse internally; petals absent or rarely with vestigial petals, yellowish, linear to narrowly lanceolate, ca. 2–2.3 × 0.2–0.4 mm, margins entire, apex acuminate, trichomes simple; ovary globose, 2.1–2.8 mm diam, trichomes stellate-porrect; styles 3, ascendant, multifid, free, with a total of 12–32 (-48) stigmatic tips, trichomes stellate-porrect; nectary disk 5-lobed, lobes rounded, glabrescent. Capsules light green to yellowish, yellow when dry, globose, 3.5–8 mm diam., unlobed, hispid surface, with stellate-porrect trichomes, slightly 3-lobed to unlobed; columella ca. four mm, flattened, without prominent lobes; seeds brown to blackish, widely ovoid to slightly globoid, 3.6–4 × 3.2–3.6 mm, smooth to slightly papillate, with cream to grayish blotches, caruncle reniform, 0.5 × 0.7–0.8 mm.

Additional specimens examined

BOLIVIA. Santa Cruz: Cordillera, al sur de Bañados del Izozog, Estancia Toborochi y Cerro Toborochi, (18°49′52″S, 62°00′55″W), 520 m, 09.I.1993, fr., G. Navarro 1728 (USZ); ibid, Cerro Toborochi, 10–12 km by abandoned gap to the SE of Estancia Toborochi, fr., 19°14″S, 62°15″W, ca. 470 m., 5–15.I.1993, I.C. Vargas et al. 1915 (WIS, USZ). BRAZIL. Bahia: Côcos, estrada Côcos–Feira da Mata, 14°15′12″S, 44°22′14″W, 484 m, fl., 09.I.2008, R.F. Souza-Silva et al. 283 (HUEFS). Feira de Santana, Distrito de Ipuaçu, 12°13′58″S, 39°04′35″W, 230 m, 05.V.2005, fr., A.P.L. Couto et al. 58 (HUEFS, HUESB). Morpará, estrada para Morpará, Beira do Rio Paramirim., 11°43′50″S, 43°13′39″W, 396 m, fl., 15.XII.2007, A.A. Conceição 2642 (HUEFS). Pilão Arcado, Brejo da Serra, 10°36′55″S 42°48′33″W, fl., 05.XII.2009, A.P. Prata et al. 1918 (MAC). Ceará: Apuiarés, Triângulo marmelheiro rasteiro, [03°94′88.89″S, 39°43′17.01″W], 09.III.2008, fr., Otilio & Chaguinha s.n. (EAC42392, HUEFS138379). Paramoti, Fazenda São João, fl., fr., 17.X.1986, E. Nunes s.n. (EAC0014856, HUEFS111097). Pentecoste, Fazenda Experimental Vale do Curu, fl., 29.I.2015, J.C. Alves s.n. (EAC57488). Espírito Santo: Santa Teresa, São João de Petrópolis, Escola Agrotécnica Federal, (19°43′20″S, 40°38′48″W, 180 m), 11.XII.1985, fr., H.Q. Boudete Fernandes 1720 (SP 2x, MO). Colatina, Rio Doce, 19°31′48″S, 40°42′02″W, 40 m, 13.II.2019, fr., F. Marinero (MBM). Minas Gerais: Jaíba, 47 km da cidade, estrada de chão sentido cidade, 15°14′13″S, 43°20′10″W, 577 m, 22.IV.2006, D.S. Carneiro-Torres 731 (HUEFS, HUESB, SP). Mato Verde, margens da rodovia Mato Verde –Monte Azul (BR 122), eight km ao norte da cidade, 15°20′07″S, 42°53′25″W, 520 m, 31.III.2004, fr., J.R. Pirani et al. 5372 (HUEFS, SP, SPF). Pará: region des Tumuc-Humac (“frontière Brésil-Guyane Française-Suriname”), 500 m SE Toukouchipan, alt. 450 m, rochers granitiques désnudés, lisière forestière, (1°59′06″N, 55°44′59″W), 19.VIII.1972, fl., fr., C. Sastre 1737 (P). Piauí: Caracol, área de entorno de lagoa, 09°12′48″S, 43°29′50″W, 533 m, 22.XI.2010, fl., E. Melo et al. 8775 (HUEFS). Castelo do Piauí, cerrado entre Castelo do Piauí e São João da Serra, (05°38′29″S, 41°72′09″W, 414 m), 26.XI.1980, fr., A. Fernandes et al. s.n. (EAC9077). São Jõao do Piauí, descida da Serra da Capivara, próximo a São Raimundo Nonato (10–11 km de Várzea Grande), (08°47′26″S, 42°28′41″W, 562 m), 05.XII.1971, fl., D. Andrade-Lima et al. 1187 (HUEFS, IPA, MAC). COLOMBIA. Cundinamarca: Municipio de San Juan de Río Seco, Corregimiento de Cambao, Finca Llandia, (4°54′18″N, 74°44′19″W), fr. 21.IX.1991, J.O. Rangel-Ch. & S. Salamanca 3222 (COL). FRENCH GUIANA. Maripasoula: Massif du Mitaraka, Crique Alama “Inselberg “Sommet colche”, 02°13′40″N, 54°28′10″W, 600 m, fr., 5.III.2015, O. Poncy 2863 (CAY, P); ibid, Grande savane roche à one km, 5 à 1″W-SW du “sommet en cloche”, (2°20′57″N, 54°47′42″W), 550 m, fr., 25.VIII.1972, J.J. de Granville 1386 (CAY, P). GUYANA. Upper Takutu-Upper Essequibo: Karanambu, secondary vegetation around the ranch, 3°45″N, 59°20″W, fr., 5.IX.1988, P.J.M. Maas 7272 (NY). MEXICO. Chiapas: Cintalapa, thorn forest near and northwest of Cintalapa along the road to colonia Francisco I. Madero, [16°32′28″N, 93°54′49″W], 29.III.1981, D.E. Breedlove 50502 (CAS); ibid, wooded slope along creek bank 27 miles west of Cintalapa, along Mexican Highway 190, [16°28′33″N, 94°03′21″W], ca. 880 m, fl., 17.V.1965, D.E. Breedlove 9966 (MICH). Jalisco: La Huerta, Rancho Cuixmala and environs. Teopa, a ranch along the Arroyo Cajones near the Puerto Vallarta –Barra de Navida Hwy, 19°25′45″N, 104°59′30″W, 30 m, 11.IX.1991, fl., E.J. Lott et al. 3726 (WIS); ibid, Chamela, Dry deciduous forest, 17.VIII.1991, fr., A. Gentry & L. Woodruff 74399 (DAV); ibid, Estación de Investigación, Experimentación y Difusión Chamela, 02.IX.1981, J.A.S. Magallanes 3098 (DAV, MEXU). Oaxaca: Santa María Huatulco, Parque Nacional Huatulco, 15°46′55″N, 96°11′58″W, 15.VI.2023, fl., C. Domínguez-Rodríguez (iNaturalist record, observation number: https://www.inaturalist.org/observations/167788472); ibid, 19.VI.2023, fl., C. Domínguez-Rodríguez (iNaturalist record, observation number: https://www.inaturalist.org/observations/168563585); ibid, 23.VI.2023, fl., C. Domínguez-Rodríguez (iNaturalist record, observation number: https://www.inaturalist.org/observations/169584486); ibid, 27.VI.2023, fl., C. Domínguez-Rodríguez (iNaturalist record, observation number: https://www.inaturalist.org/observations/174466881); ibid, 15°46′51″N, 96°11′63″W, 25.VI.2023, fl., E.S.V. Arellanes (iNaturalist record, observation number: https://www.inaturalist.org/observations/169780836). SURINAME. Sipaliwini: Paka paka, granitic plateau near km ±8 in line from Paka paka (Saramacca R.) to Ebbatop (v. Asch v. Wijcks-Range), 9.II.1951, P.A. Florschüts 1291 (NY); ibid, lower slopes of Voltzberg, 04°41″N, 56°11″W, 150–175 m, 02.VIII.1979, fl., fr., G.L. Webster 24136 (DAV, MO, TEX, WIS); ibid, Central Suriname Nature Reserve, Voltzberg top I, 4°41″N, 56°11″W, 150–245 m, 13.IV.2002, fr., A. Gröger et al. 1300 (U); ibid, op helling van Voltzberg, op humus re op rease rots, Boven-Coppename, 22.IX.1954, fl., A.M.W. Mennega 62 (U); ibid, Voltzberg, 19.IX.1993, J. Lanjouw 902 (BR, K, L, NY, P, RB, U); ibid, halfway Voltzberg, 19.IX.1993, J. Lanjouw 899 (K, US). VENEZUELA. Carabobo: Road from San Diego to Valencia, XIII.1941, J. Saer 816 (US); Entre Las Trincheiras y La Entrada, (10°18′06″N, 68°04′10″W), fl., 2.V.1920, H. Pittier 8821 (VEN). Distrito Capital: Puerto Escondido, on dry slopes, (10°24′33″N, 66°55′25″W), 18.IV.1930, H.F. Pittier 13421 (F); El Valle and Laguna de Espino, 18.VI.1891, Eggers 13143 (US). Lara: Dtto. Palavecino. Parque Nacional Terepaima, camino subiendo desde La Mata, en las lomas de la parte este, (10°00′36″N, 69°16′58″W), 700 m, fr., 10.VI.1979, C. Burandt Jr. V0809 (BRIT). Mérida: Entre Estanques y Puente de La Victoria, (8°26′28″N, 71°34′12″W), 700 m, 7.V.1953, Bernardi 498 (P). Miranda: Caracas, reforested hills of the Caracas Botanical Garden, 980 m, 10.XI.1974, P.E. Berry 99 (VEN); ibid, Bosque deciduo secundario y perturbado entre las quebradas afluentes al Río Guarita, al sur del Cementerio Monumental del Este, 1,000 m, 17.VIII.1975, fr., J.A. Steyermark & P.E. Berry 112078 (F, NY, VEN); ibid, J.A. Steyermark & P.E. Berry 112082 (NY); Guaicaipuro, Sector Begonia-Caobal, Parcela Loma Brisa. Bosque seco en laderas que dan al NE, 10°16′36.57″N, 66°56′30.27″W, 785 m, 23.VII.2009, R. Riina & C. Reyes 1843 (MA). Monagas: Roadside between Caicara and San Félix, 7.VI.1967, fl., R.A. Purcell 9220 (NY). Táchira: inmediaciones de San Cristóbal, (7°40′19″N, 72o13′47″W), 27.X.1953, L. Croizat s.n. (COL56784).

Distribution, phenology, conservation status, and ecological niche modeling

Croton anomalus was here found to be a relatively widespread neotropical species occurring in fragments of dry forest (SDTF) and savanna vegetation in Bolivia, Brazil, Colombia, French Guiana, Guyana, Mexico, Suriname, and Venezuela (Fig. 4, Table 5). This increase in the geographic range of C. anomalus compared to the previous knowledge (known only from Venezuela) is the result of the synonymization of C. acapulcensis, C. chiapensis, and C. stahelianus, as well as by the numerous new records discovered in Bolivia, Brazil, and the eastern Guiana Shield region. Records of C. anomalus were found primarily in SDTFs, and secondarily in transition areas of STDF with Savanna (Cerrado phytogeographic domain, and Guianan savannas) or Tropical Rainforest (Fig. 4). The species altitudinal range varied from 30 to 1,100 m. Some herbarium labels report that the species grows in sandy and clay soils, or on rocky outcrops (inselbergs in the Guianas). Croton anomalus was most frequently found inside forest vegetation, growing in shade and glade zones. Although restricted to the Caatinga domain, C. blanchetianus and C. jacobinensis were widely distributed within this domain, occurring in open areas (C. blanchetianus) or transition zones to montane and Atlantic forests (C. jacobinensis) (Fig. 5, Table 5).

Figure 4 Distribution map of Croton anomalus.

Red star: type locality; black circles: previous records of C. anomalus; blue circles: new records and previous records of synonyms. Records from Mexico, Suriname and northernmost Brazil are based on the synonymizations proposed here. Shapefiles were downloaded from http://www.efrainmaps.es. (Americas), http://www.dryflor.info/data/datasets (Dry Forests), and Olson et al. (2001) https://worldwildlife.org/publications/terrestrial-ecoregions-of-the-world (Additional ecoregions).

Table 5 Morphological comparison between the species included in the Croton anomalus group.

Character	Croton anomalus	Croton blanchetianus	Croton jacobinensis (=C. sonderianus)	
Indumentum	Stellate to multiradiate, porrect	Stellate, sublepidote to lepidote	Stellate to multiradiate	
Leaf blade	Ovate to elliptic, margins entire, irregularly sinuate or slightly serrate	Ovate to ovate-lanceolate, elliptic, margins entire	Cordiform, margins entire	
Venation	Eucamptodromous to brochidodromous	Eucamptodromous	Actinodromous	
Inflorescences	Unisexual (staminate) or bisexual (1–5 cm long)	Bisexual (3–12 cm long)	Bisexual (5–18 cm long)	
Number of pistillate sepals	5–6 (7)	5	5	
Inner surface of pistillate sepals	Sparsely hirsute, trichomes stellate, porrect	Glabrous	Densely velutinous, trichomes stellate to multiradiate	
Styles	Ascending, free	Ascending, united forming a small column	Patent or ascending, free (rare slightly united at the base)	
Capsule trichomes	Stellate to multiradiate, porrect	Lepidote, rarely sublepidote	Stellate to multiradiate	
Seed surface	Smooth, slightly papillose	Smooth	Rugose	
Geographic distribution	SDTFs of Bolivia, Brazil, Mexico, Suriname and Venezuela	Caatinga SDTF (Brazil)	Caatinga SDTF (Brazil)	

Figure 5 Distribution of Croton blanchetianus and C. jacobinensis.

Distribution map of (A) Croton blanchetianus (red circles) and (B) Croton jacobinensis (black circles) in the Caatinga phytogeographic domain. Records outside Caatinga are those within transition zones with Atlantic Forest.

Regarding phenology, Croton anomalus has been recorded flowering and fruiting in January (Bolivia), April to November (Venezuela), May and August (French Guyana), June to August (Mexico, Suriname, Venezuela), September (Colombia), and from October to May (Brazil).

Based on its EOO of 14,018,878.897 km2 and AOO of 232,000 km2, we tentatively classified C. anomalus as Least Concern (LC). This species was found in some protected areas, such as the Henri Pittier and Terepaima National parks (Venezuela), Parque Nacional del Gran Chaco Kaa-lya (Bolivia), Natuurreservaat van Centraal-Suriname (Suriname), and Parque Nacional Serra das Confusões, Parque Nacional Serra da Capivara, Parque do Tumucumaque, and Reserva Particular do Patrimônio Natural Mãe-da-Lua (Brazil).

The average values of AUC in the three replicate runs were above 0.9 for all species, with standard deviations ≤0.023 (Table 6). The scores demonstrate the high capacity of the model in precisely discriminating suitable areas for the occurrence of species, and therefore it represents a good performance in our analysis. Annual precipitation (bio12) and the precipitation of the warmest quarter (bio18) were the bioclimatic variables found to be important for the distribution of all three species (Table 6). When comparing the most important variables for each species individually, bio6 was the most important for C. anomalus while bio12 and bio4 were the most important for C. blanchetianus and C. jacobinensis, respectively. The temperature seasonality (bio4) was scored as important only for C. jacobinensis. The suitability of areas of occurrence (Fig. 6) followed the distribution pattern of the SDTFs for C. anomalus and overlapped with the majority of the areas where the three species have been recorded (only Caatinga SDTF for C. blanchetianus and C. jacobinensis). Nonetheless, the Caribbean region and the coastal region of Ecuador and Peru were indicated as moderately suitable areas for C. anomalus (Fig. 6), as was northern South America for C. jacobinensis, even if there are no occurrence records of these species in those areas, respectively (Figs. 4 and 5).

Table 6 Descriptive statistics of the Ecological Niche modeling for the three studied Croton species.

Species	AUC (SD ±)	Variables mostly contributing (PC [%]/PI)	
C. anomalus	0.936 (0.023)	bio6 (25.3/45.5), bio12 (20.1/19), bio03 (9.2/0.3), bio18 (8.8/3.6)	
C. blanchetianus	0.972 (0.004)	bio12 (22.1/53.9), bio18 (22/3.9), bio6 (15.4/1.4), bio17 (14.7/0.2)	
C. jacobinensis	0.961 (0.013)	bio4 (23.6/44.8), bio12 (16.2/23.2), bio18 (13.2/2.2), bio17 (12.6/8.8)	
Notes.

AUC Area under the curve

SD Standard deviation

PC Percent contribution of the variables

PI Permutation importance

Figure 6 Modeled distribution for species recognized in the Croton anomalus group.

(A) Croton anomalus, (B) C. blanchetianus, and (C) C. jacobinensis. The scores (0–1) in the scale represent the point-wise mean of probability of occurrence in the three replicates run. Hotter colors: high suitability for occurrence; Cooler colors: less suitability for occurrence. Values > 0.66 were considered more likely to be suitable .

Discussion

Typification and new synonyms

Croton anomalus was described based on a collection of Henri F. Pittier, from Venezuela, with Pittier 11757 indicated as the type material. The author (Pittier, 1930) did not mention where the type material was deposited nor if there was only one specimen (automatically to be considered the holotype [Art. 9.1 of the ICN, Turland et al., 2018]). Three specimens of Pittier 11757 are found in herbaria A, US, and VEN, and are considered syntypes (Art. 9.6 of the ICN, Turland et al., 2018). Since the name C. anomalus was published based on syntypes, any lectotype must be selected from among them (Art.9.11 and 9.12 of the ICN, Turland et al., 2018). The specimen in best condition is A00047223, and it is designated here as the lectotype of C. anomalus.

Croton stahelianus was described by Lanjouw (1931) based on specimens from Suriname collected by Gerold Stahel n. 611. According to the Euphorbiaceae of Surinam index (Lanjouw, 1931), when ‘B.W.’ is indicated in the protologue it means that the material was collected by the ‘Boschwesen’ (Forestry Department), with which Stahel was involved at the time (Stafleu & Cowan, 1985). So, B.W. n. 7002 represents the collection Stahel n. 611, as indicated in the labels of the type material. There are two specimens of Stahel n.611 deposited in the U herbarium, of which we selected U0178808 as the lectotype of C. stahelianus. Lanjouw (1931) did not assign C. stahelianus to any infrageneric group, but described the species as related to C. doctoris S.Moore and C. flavens var. flavens Müll.Arg., both representatives of C. section Adenophylli Griseb (Van Ee, Riina & Berry, 2011). Both aforementioned species can be distinguished from C. stahelianus for their acropetiolar glands, bifid styles and columella with prominent apex (all absent in C. stahelianus). The morphological characteristics included in the protologue of C. stahelianus that are discordant with those of C. anomalus are: axillary or terminal inflorescences (vs. only terminal on C. anomalus), 11 stamens (vs. 16 on C. anomalus), pistillate flowers with five sepals of glabrous inner surface (vs. 6–7 sepals of villous inner surface on C. anomalus).

The original description of Croton chiapensis is somewhat controversial because some character states described by Lundell (1942) are not seen in the type collection, namely staminate flower with six sepals and six petals, and pistillate flowers with bifid styles. The staminate flowers are pentamerous and the styles are 4-fid in the few open pistillate flowers (the inflorescences mostly have pistillate flower buds). Also, other authors have indicated five sepals and petals for the staminate flowers (Pittier, 1930; Gordillo & Ramírez, 1990). Based on this, we believe that Lundell (1942) confused the number of sepals in staminate flowers with those in the pistillate ones (5–6) when describing C. chiapensis. The only characters in C. chiapensis differing from C. anomalus, based on the protologue, are the presence of 18 stamens (vs. 14–17 in C. anomalus), and the bifid styles.

Gordillo & Ramírez (1990), in their description of C. acapulcensis, mentioned only C. alamosanus Rose as its closest relative. These authors used traits of leaves, stipules, and flowers to differentiate the two species. However, our review of the protologues and type specimens of species in Croton sect. Lasiogyne shows a strong morphological resemblance between C. acapulcensis and C. anomalus from Venezuela, prompting us to further study and compare these two taxa. According to the protologues, C. anomalus differs from C. acapulcensis by the ovate to ovate-lanceolate leaf blades (vs. oblong-lanceolate in C. acapulcensis), irregularly sinuate or dentate to serrate leaf margins (vs. entire), linear to lanceolate stipules (vs. subulate), inflorescences of 5–7 cm long (vs. 2.5–5 cm long), 16 stamens (vs. 14–15), and the pistillate flowers with six to sometimes seven sepals (vs. five or sometimes six) (Table 5, Fig. 7).

Figure 7 Morphological comparison of species in the Croton anomalus group.

(A–C) Croton anomalus, (D–F) C. blanchetianus, and (G–I) C. jacobinensis. (A, D, G) General aspects of leaves and inflorescences, (B, E, H) pistillate flowers of five and (B , upper right) six sepals, and (C, F, I) capsular fruit. Photographs: (A–B) (A. S. Farias-Castro), reproduced with permission.

After examining numerous specimens from Brazil, Bolivia, Colombia, Mexico, Suriname, Guyana, French Guiana, and Venezuela (including the type collections), we determine that the characteristics listed above overlap within and among populations. Character states such as entire or serrate leaf blade margins are found on the same individual (even on the same branch). The presence of pistillate flowers with seven sepals (probably responsible for the epithet “anomalus”) is a rare state for this species, although not so abnormal in the genus, where as many as 10 sepals have been observed in C. sincorensis Mart. ex Müll. Arg. (Sodré et al., 2019). Thus, we find weak morphological evidence to sustain C. acapulcensis, C. anomalus, C. chiapensis and C. stahelianus as four distinct taxonomic entities. As shown by the morphological data, C. acapulcensis, C. chiapensis, and C. stahelianus are better positioned as synonyms of C. anomalus. Additionally, the phylogenetic reconstruction also agrees with the morphological evidence where the two specimens of C. acapulcensis and C. stahelianus are intermingled in a highly supported clade along with C. anomalus specimens representing most of the geographic range of the species. As noted above, we were unable to obtain DNA sequence data for C. chiapensis (Fig. 1), so the synonymization of this name is based on evidence other than molecular phylogenetics (i.e., morphology, ecology, geographic distribution data).

Systematics of the Croton anomalus group

We conducted the most complete phylogenetic analysis of C. sect. Lasiogyne to date, even if it only includes 10 of the more than 40 species currently assigned to the section (Fig. 1). Also, the phylogenetic position of C. anomalus and C. blanchetianus as members of section Lasiogyne was demonstrated here for the first time (Fig. 1). All the sampled species had their phylogenetic position congruent with previous phylogenetic analyses of Croton (Berry et al., 2005; Van Ee, Riina & Berry, 2011; Arévalo et al., 2017). In contrast with Van Ee, Riina & Berry (2011), section Lasiogyne is strongly supported as monophyletic (PP = 0.95; Fig. 1), being more closely related to section Julocroton rather than to section Heptallon as in Van Ee, Riina & Berry (2011). The latter section is in turn sister to the clade formed by sections Lasiogyne and Julocroton (Fig. 1). However, the monophyletic status of section Lasiogyne remains to be confirmed by the inclusion of a more comprehensive taxon sampling in future phylogenetic analyses. Our results are in accordance with previous taxonomic classifications based on morphology, which placed C. anomalus and C. blanchetianus as members of C. sect. Lasiogyne (e.g., Van Ee, Riina & Berry, 2011; Rossine et al., 2023).

Based on our extensive taxon sampling and emended morphological description (capsules and seeds were not described in the protologues of C. anomalus and C. acapulcensis [Pittier, 1930; Gordillo & Ramírez, 1990]), C. anomalus is definitely a member of section Lasiogyne, as indicated by Van Ee, Riina & Berry (2011), who based this placement on morphology alone. This species is characterized by a set of character states presented in species of section Lasiogyne (Van Ee, Riina & Berry, 2011; Rossine et al., 2023), specifically a stellate indumentum, lack of nectary glands in leaves, bracts and sepals, reduplicate-valvate pistillate sepals, and multifid styles.

Morphologically, C. anomalus is most similar to the Brazilian C. blanchetianus (Fig. 7; Table 5). The two species share the monopodial branching, lanceolate stipules, general aspects of the leaves such as the stellate indumentum (making the leaf blade light green on the abaxial surface), ovate leaf blades, entire leaf margins, eucamptodromous venation, multifid styles, and smooth seeds. Both species are morphologically variable and plastic, showing some overlap in character states among species. For example, in C. blanchetianus the stipule shape is usually lanceolate, but can also be reniform or auriculate; and the leaf blade, which is usually ovate, can also be cordiform (Rossine et al., 2023). On the other hand, the entire leaf margin can be irregularly sinuate to slightly serrate, and the venation can be brochidodromous instead of eucamptodromous (the most frequent state) in C. anomalus.

Croton anomalus and C. jacobinensis share the shrubby habit, stellate to multiradiate indumentum, lack of nectary glands at the apex of the petiole, lanceolate stipules, leaf blade coloration, entire leaf margins, linear to 3-lobed bracts, 14–17 stamens, reduplicate-valvate sepals in pistillate flowers, and multifid, free styles (Rossine et al., 2023). Again, some character states can be polymorphic in C. anomalus (e.g., irregularly sinuate to slightly serrated leaf margins, and brochidodromous venation). The main differences between C. anomalus, C. blanchetianus, and C. jacobinensis are indicated in Table 5 and Fig. 7, as well as in the key below.

Identification key to Croton anomalus and close relatives with similar morphology (Croton anomalus group)

1. Leaf blade always cordiform; venation actinodromous; styles patent; seeds rugose ...................................................................... Croton jacobinensis (=C. sonderianus)	
1′. Leaf blade ovate to oval-lanceolate, rarely cordiform in C. blanchetianus; venation eucamptodromous to brochidodromous; styles ascending; seeds smooth ........................................................................................................................ 2	
2. Stipules always lanceolate; leaf margin entire, irregularly sinuate or slightly serrate; inflorescences unisexual (staminate) or bisexual; pistillate flowers with 5–6 (7) sepals, slightly unequal in size, with stellate-porrect trichomes internally; styles free; nectary disk 5-lobed; stellate to multiradiate, porrect, trichomes on the capsule; species distributed in dry forests and Guianan savanna vegetation ........................................................................................................Croton anomalus	
2′. Stipules lanceolate, reniform or auriculate; leaf margin always entire; inflorescences always bisexual; pistillate flowers with 5 sepals, equal in size, glabrous internally; styles united at the base forming a short column; nectary disk with 5 free segments; lepidote to sublepidote trichomes on the capsule; species endemic to Caatinga dry forest ....................................................................................................... Croton blanchetianus	

Species distributions and ecological niche modeling

Croton anomalus as newly circumscribed, is reported to occur along most of the fragments of Neotropical STDF, with many new records in the Caatinga phytogeographic domain (Brazil) and Gran Chaco region (Bolivia). The previously known records of this species were restricted to Venezuela, in six states: Carabobo, Distrito Federal, Lara, Nueva Esparta, Yaracuy, and Zulia (Pittier, 1930; Hokche, Berry & Huber, 2008). Hokche, Berry & Huber (2008) do not provide examined herbarium material for C. anomalus, so we revisited all the Venezuelan specimens identified as C. anomalus available. We found specimens from Nueva Esparta (H. Gines 3125 [US]) and Zulia (G.S. Bunting 11827 [NY]) that were misidentified as C. anomalus, so presumably Hokche, Berry & Huber (2008) used these erroneous specimens in their distribution data. We were not able to find any specimens determined as C. anomalus from the Yaracuy state. Although there are not herbarium records of C. anomalus from Zulia, Nueva Esparta, and Yaracuy, the species could occur in these three Venezuelan states due to their floristic similarity with neighboring areas where we found well identified records (Fig. 4). The occurrences of C. anomalus in Mexico are based on the synonymization of C. acapulcensis, C. chiapensis, and some other iNaturalist records, while the ones from the eastern Guiana Shield (Brazil, French Guiana, Guyana, Suriname) are based on the synonymization of C. stahelianus. Finally, C. blanchetianus and C. jacobinensis are well-documented species, particularly in terms of their geographic distribution, as described in studies dealing with Croton of the Brazilian Caatinga domain (e.g., Silva et al., 2010; Oliveira et al., 2023; Rossine et al., 2023).

Most populations of C. anomalus are found in SDTF of Bolivia, Colombia, Mexico, Venezuela, and Southern Brazil, while the specimens of the eastern Guiana Shield are found in open secondary vegetation or dry forests associated to inselbergs (Guianan savannas). According to DRYFLOR et al. (2016), the Caatinga STDF shares less than a hundred species with the Mexican SDTF, and about 100–240 species with the northern South American nuclei of SDTF. Croton anomalus can now be added to this group of species shared between distant SDTF across the Neotropics.

It is noteworthy that of the potential areas with high suitability (>0.69, Fig. 6A) for the occurrence of C. anomalus, five zones (Central America, the Caribbean, Ecuador, the Galapagos Islands, and Peru) have no collection records of this species. This may be due to gaps in knowledge of the local SDTF floras in these zones. In fact, new records and new species of dry forest taxa have been continuously published in the last decade, including several Croton species (e.g., Cornejo, 2017; Feio et al., 2018; Zapata & Villarroel, 2019; Marques et al., 2020; Mateo-Ramírez & Riina, 2020; Hammel & Arias, 2022; Villanueva-Tamayo et al., 2023; Villanueva-Tamayo & Aymard-Corredor, 2024; Martín-Muñoz et al., 2024). An alternative explanation is the high degree of isolation of some these suitable areas (Ecuador and Peru), i.e., the existence of large areas of low-climatic suitability separating them from the rest of suitable areas where C. anomalus is present in South America (Fig. 6A). Likewise, in the case of the Galapagos Islands and the Caribbean Islands, the ocean can also act as a physical barrier for seed dispersal. More difficult to explain is the absence of C. anomalus in the suitable areas identified by the model in Central America. Colombia, Venezuela and Central America share a significant proportion of their STDF species (DRYFLOR et al., 2016), so it can be speculated that for a species distributed from Mexico to Bolivia and Brazil, its absence in Central America is probably due to gaps in the floristic knowledge of SDTF.

The niche analysis of Croton jacobinensis also shows suitable areas in northern South America (coastal of Venezuela and Ecuador) where this species is not present (Fig. 6C). In this case, the Amazon rainforest could represent a strong geographic and climatic barrier for the south-to-north direction of dispersal of C. jacobinensis, since populations of this species are found in all other suitable areas south of the Amazon region.

A better sampled phylogeny of Croton sect. Lasiogyne along with a biogeographic analysis, as has been done in other Neotropical STDF plant groups (Lavor et al., 2019; Silva, Riina & Cordeiro, 2020; Hurbath et al., 2021; Pezzini et al., 2021), will allow us to take evolutionary time into consideration for a better understanding of the current distribution of the studied species as well as other Croton species restricted to STDF fragments throughout the Neotropics.

Impacts of misidentification in herbaria

The case of Croton anomalus highlights the importance of updating taxonomic revisions in the Neotropics, especially in plant groups occurring in fragmented biomes such as the SDTF. It also shows the effect of specimen misidentifications on biogeographic knowledge and consequently on species conservation assessments. Many specimens of Croton anomalus were found in Brazil misidentified as C. sonderianus (a synonym of C. jacobinensis) or as C. blanchetianus. In fact, these species are very similar morphologically, but we have demonstrated that there are morphological characters distinguishing them, which are also supported by strong phylogenetic divergence. Morphological characters and/or ITS sequences can be used to correctly identify specimens of this group, even those with populations overlapping in the same biogeographic region such as the Brazilian Caatinga. Croton anomalus was also found in Bolivia and Suriname identified only at the genus level (as Croton sp.), whereas in Mexico it has been identified as C. acapulcensis and C. chiapensis (proposed here as new synonyms) and in French Guyana, Guyana, and Suriname as C. stahelianus (also a new synonym proposed here).

Brazil is the country where most of the taxonomic problems related to C. anomalus have arisen. Specimens of either C. blanchetianus or C. jacobinensis have been routinely identified as C. sonderianus in herbaria over the decades. The use of the name C. sonderianus to treat specimens of C. anomalus, C. blanchetianus and C. jacobinensis seems to have been popularized since the 1980s due to misidentifications by Euphorbiaceae experts, which were followed subsequently by local taxonomists. Also, it seemed that the specimens were not identified by comparing them with the protologue and type collection of C. sonderianus. The lack of knowledge of the presence of C. anomalus in Brazil has caused divergence even among Croton experts due to the close morphological affinities among these species in question. The synonymization of C. sonderianus under C. jacobinensis Gomes, Sales & Melo (2010) is reinforced here, and we recommend avoiding the use of C. sonderianus other than as a synonym of C. jacobinensis. The various sources of evidence shown here support the recognition of three species in the Croton anomalus group: C. anomalus, C. blanchetianus, and C. jacobinensis.

Supplemental Information

Supplemental Information 1 Sequences of ITS (fasta format) used in the study of Croton anomalus group

Supplemental Information 2 Sequences of trnLF (fasta format) used in the study of Croton anomalus group

Supplemental Information 3 Phylogenetic reconstruction of the Croton anomalus group based on ITS data

Phylogenetic reconstruction of the Croton anomalus group illustrated by a majority consensus tree obtained from the Bayesian analysis of the ITS dataset. Names in bold are those newly generated in this study.

Supplemental Information 4 Phylogenetic reconstruction of the Croton anomalus group based on trnLF data

Phylogenetic reconstruction of the Croton anomalus group illustrated by a majority consensus tree obtained from the Bayesian analysis of the trnL-F dataset. Names in bold are those newly generated in this study.

We appreciate the access to the collections and the information provided by the curators and staff of all the herbaria cited. We thank Regina Carvalho for the line drawings; Daniela Carneiro-Torres and Wesley Sá for reviewing a previous version of this manuscript. We thank the Real Jardín Botánico systematics laboratory for the use of its facilities and materials, Mónica García-Gallo Pinto for her help with DNA extractions and Yolanda Turégano Carrasco for helping with the phylogenetic analyses and lab issues. Photographs were kindly provided by their authors (cited in figure captions). A special thanks to Hermann Redies (in memoriam) for sharing information and pictures of Croton anomalus, leading us to the development of this research. We thank the three reviewers, P.E. Berry, V. Steinmann, and G. Levin, for their useful suggestions and corrections to an earlier version of this article.

Additional Information and Declarations

Competing Interests

Author Contributions

DNA Deposition

Data Availability

The authors declare there are no competing interests.

Yuri Rossine conceived and designed the experiments, performed the experiments, analyzed the data, prepared figures and/or tables, authored or reviewed drafts of the article, and approved the final draft.

Ricarda Riina conceived and designed the experiments, performed the experiments, analyzed the data, prepared figures and/or tables, authored or reviewed drafts of the article, and approved the final draft.

Otávio L.M. Silva performed the experiments, analyzed the data, prepared figures and/or tables, authored or reviewed drafts of the article, and approved the final draft.

Rafael Louzada performed the experiments, authored or reviewed drafts of the article, and approved the final draft.

The following information was supplied regarding the deposition of DNA sequences:

The sequences are available at GenBank (Table 1):

KF208629.1, EU477862.1, PQ350252, PQ350253, PQ350254, PQ350256, PQ350257, PQ350255, PQ350258, EU478094.1, HM071943.1, HM564076.1, EU586901.1, PQ350259, PQ350260, PQ350261, PQ350262, MW263138.1, AY971208.1, PQ350263, HM564081.1, EU478068.1, FJ614722.1, HM044795.1, PQ350264, PQ350265, EU478108.1, HM564087.1, EU478096.1, EF421789.1, FJ614766.1, HM044802.1, EF421792.1, PQ350266, EF421752.1, EU477882.1, DQ227537.1, KF208632.1, EU478122.1, PQ458518, PQ458519, PQ458520, PQ458522, PQ458523, PQ458521, PQ458524, EU497702.1, HM071965.1, HM564213.1, EU586955.1, PQ458525, PQ458526, PQ458527, PQ458528, MW266678.1, AY971297.1, PQ458529, HM564217.1, EU478159.1, FJ614783.1, HM044776.1, PQ458530, PQ458531, FJ614786.1, HM564222.1, EU497720.1, EF408131.1, FJ614804.1, HM044782.1, EF408133.1, PQ458532, EF408125.1, EU478125.1, DQ227569.1.

The following information was supplied regarding data availability:

The aligned data matrices are available in the Supplementary Files.

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
