# Peer review of "Untangling the taxonomic knot of Croton anomalus (Euphorbiaceae), a Neotropical dry forest shrub"

_PeerJ, doi:10.7717/peerj.19176_

## Round 0.1 · original submission · Minor Revisions

Dear Dr. Riina,

The reviewers found the manuscript to be convincing and based on solid data. They made a number of suggestions to further improve the quality of the manuscript. They also made several suggestions to improve the English language and sentence formulation which is insufficient in some parts. Please, respond to these points. Once again, thank you for submitting your manuscript to PeerJ and we look forward to receiving your revision.

Sincerely,
Gabriele Casazza

·

Basic reporting

There are quite a few deficiencies in the English which can be improved. Some assertions are questioned concerning the title, the 'forgetting and ignoring' of Croton anomalus, and the 'rediscovery' of the species in Brazil and other areas.

Experimental design

No comment

Validity of the findings

The morphological and molecular results are convincing, but the ecological niche modeling less so. For the latter, I think the results are more nuanced than is briefly discussed in the text, and they need further elaboration.

Additional comments

I feel that the authors have done a credible job of recircumscribing Croton anomalus, a species of Croton from dry forested areas of Mexico and South America, and discussing its relevance to the distribution of Seasonally Tropical Dry Forests (STDFs). It is nice to see a combination of morphological and molecular evidence used to support their conclusions. On the other hand, I don’t feel that the results of the niche modeling exercise (Fig. 6) were fully explained or interpreted. By this I mean that it would be necessary to give a better explanation of how to interpret the scores on the scale from 0 to 1 and their appearance on the maps (e.g., are these the point-wise means mentioned in the legend?). On the surface, I would interpret the hotter colors to indicate higher probability of occurrence and the cooler colors less probability, but overall there is at least some coloration in most of the STDF areas shown in the Neotropics for each of the three species. Also, the modeling of the likelihood of occurrence within the Caatinga domain in Brazil is nearly identical for the three species, yet the actual distribution maps (Figures 4 and 5) show that Croton anomalus occurs mainly on the outer fringes of the Caatinga, whereas C. blanchetianus and C. jacobinensis are more evenly distributed across that area. That seems to me like an important distinction that should be noted and discussed.

Most of my comments appear on a track-change and commented version of the manuscript in Word, which I can hopefully attach to the review {Note to Editor and Authors: the review system does not allow me to attach a Word doc, and when it gets changed into a pdf, it loses all the edits and some comments are misplaced, so some other mechanism is needed for me to pass along the version in Word}. I had issues with a number of assertions in the manuscript that I do not believe are accurate. For instance, beginning with the Abstract, “Croton anomalus has been forgotten and ignored for almost a century, causing a series of taxonomic confusions.” That is not true, given that many of the same specimens that are cited in the text from Venezuela were previously identified as Croton anomalus, and the species and its known distribution in Venezuela in six different states were treated in the Nuevo Catalogo de la Flora Vascular de Venezuela (Hokche et al., 2008). In the next sentence they state that C. anomalus was “rediscovered in other countries, where it has been confused with several species…”. To me, that does not qualify as a rediscovery but rather as a new circumscription of C. anomalus once those misidentifications were recognized and corrected. Likewise, it is stated that Croton chiapensis is only known from the type specimen (Matuda 2614), yet the Breedlove 50502 specimen from Chiapas had also been previously identified to that name. Even the proposed title of the paper seems a bit overdone to me (after all, it is just one “taxonomic knot”, and I would not assess the case of Croton anomalus as particularly “intricate”; lastly, I don’t know why C. anomalus would be considered a “specialist” of the seasonally dry tropical forests, it just largely occurs in that biome). I offer a suggestion for different wording in the annotated version.

In the taxonomic treatment, I would caution against adding geographical coordinates that were not provided by the original collectors on their labels. The first example is the type of Croton anomalus itself, which never provided coordinates. Perhaps the coordinates listed were inferred by the location name, but if so, then they should be entered in brackets [...] to indicate the inference. The second example is on one of the additional specimens examined, namely Pittier 13421 (F), which gives both erroneous coordinates and elevation (these were never given on the specimen labels, and they place the specimen in the wrong state, at the wrong elevation and in a totally different, moister biome). If that information was used in the niche modeling exercise, then it could obviously introduce errors into the results.

There are sections of the Discussion, particularly the section on “Species distributions and ecological niche modeling,” where I could not easily follow the thread of the discussion, notably from line 444 to 448. A series of references are cited on “recent biogeographic studies”, but without sufficient information to give them context or relevance to Croton anomalus. The next sentence states: “When including species present in more than two distinct fragments, they often present corridors connecting those regions.” Although a reference to Colli-Silva is given, I didn’t go to look it up to see if it clarified the quoted phrase, but it just seemed to come out of nowhere in that part of the discussion, without much connection to the discussion or follow-up.

Hopefully my edits on the manuscript provide additional suggestions for improving the wording in several instances, and for filling in details on the type specimens of Croton stahelianus at the Naturalis Herbarium of the Netherlands. Aside from Table 6 (that is, in addition to it), I feel that it is essential that the authors provide a classical key to differentiate Croton anomalus from C. blanchetianus and C. jacobinensis. Since they mention considerable overlap and variation in certain characters among these species, that is where a well-crafted key can help clarify the distinctions among them.

·

Basic reporting

This manuscript is well-written. I have made a few suggestions for improvements on the pdf version. The relevant literature has been cited and the context for the work is explained clearly and completely. The structure of the paper is standard. The figures are excellent, all are important, and no additional ones are needed. I do suggest that the caption for Figure 5 be expanded to explain that the colors represent probability that the area is suitable for the species’ modeled habitat requirements. The tables are appropriate, clear, and complete. The raw data are all accessible. I am supportive of presenting only the tree based on the concatenated data in the paper itself, with the trees based on the individual gene regions presented only as supplementary material.

Experimental design

The work presented here fits within the aims and scope of the journal. The hypotheses reflect the underlying taxonomic problems and are clearly stated. The combination of morphological comparisons, phylogenetic analysis of DNA-sequence data from appropriate regions, and niche modeling provides a powerful approach to testing the hypotheses. Sampling is well designed and sufficient. The methods are clearly described.

Validity of the findings

The results clearly support the conclusion that three species are represented among the material sampled, with the synonymy as listed. The underlying data are provided and robust. The conclusions are clearly stated, address the hypotheses, and are all supported by the results.

Additional comments

This paper can serve as a good model for examining species circumscriptions and relationships within taxonomically difficult groups.

·

Basic reporting

I like this article. It is well-written and resolves a taxonomic problem involving various species and names within the giant genus Croton. The authors cite the relevant literature, and the figures and tables are pertinent and well prepared. It certainly meets your journal's standards.

Experimental design

The methods are appropriate for the study.

Validity of the findings

The results and discussion are properly presented and will improve our understanding of neotropical plant diversity.

Additional comments

The authors use the term “Croton anomalus group.” I think that the words Croton anomalus should also be italicized. Also, how is the group defined? It seems only to be a group of three species that have been previously confused, and the results clearly demonstrate that these do not form a monophyletic group. Are there morphological features or a combination of such features that unite these species but separate them from other members of Croton.

In lines 119–120, the authors state that they have consulted herbarium online data portals. I suggest that they include a list of which herbaria were consulted online.

In line 388, why isn’t Croton chiapensis also included? Doesn’t the morphological data also suggest that it is a synonym of C. anomalus?

Is there a reason to use different formats for dates of the type specimens and the additional specimens examined (p.e., Jan. 6, 1994 vs. 6.I.1994)?

The authors state that Croton chiapensis is known only from the type, but then they cite a collection from Chiapas (D.E. Breedlove 50502) that likely corresponds to the concept of Croton chiapensis. Why do they not consider this to represent Croton chiapensis?

I don’t think that it is necessary to include the abbreviation “a.s.l.” after the abbreviation for meters. It is usually not included in English.

The authors start their paper with a discussion of cryptic or very similar species. However, in the case of the species involved here, they are clearly not cryptic, and as evidenced by their results, they don’t seem that similar. It just looks like a case in which nobody had previously done the necessary work to understand their delimitation.

There are many observations of Croton anomalus in the platform iNaturalist (https://www.inaturalist.org/observations?taxon_id=1475782). These are from the state of Oaxaca, Mexico, which is not included by the authors of this paper in the distribution of the species. These should be examined, and if the authors agree that they belong to the species, they should be mentioned and considered in the distribution map.

A few minor grammatical suggestions and corrections:
Lines 26–27 change “this species is here rediscovered in other countries” to “this species is here reported from other countries”
Line 58: change “in taxonomy, and is a common problem in” to “in taxonomy and is a common problem in”
Line 59: change “assessing biological conservation” to “assessing conservation status”
Line 91: change “century ago (Pittier, 1930) and it was known” to “century ago (Pittier, 1930), and it was known”
Line 95: change “known” to “considered”
Line 125: change “known from the type” to “known from the type collected more that 85 years ago”
Line 160: change “recommendations” to “rules and recommendations
Line 191: change “region ITS” to “ITS region”
Line 287: change “Exp’r”entación” to “Experimentación”
Line 310: change “disjunctively” to “disjunctly”
Line 315: change “of elevation” to “in elevation”
Line 317: change “shadow” to “shade”
Line 324: change “national parks, in Venezuela” to “national parks in Venezuela”
Lines 345–346: change “is A00047223 and it” to “is A00047223, and it”
Line 379: change “(vs. 5 or sometimes 6 in C. acapulcensis)” to “(vs. 5 or sometimes 6)” [note: as in the previous features, it is understood that the values in parentheses are those of C. acapulcensis.]
Line 385: change “where even 10 sepals” to “where as many as 10 sepals”
Lines 497–498: change “is reinforced and we recommend” to “is reinforced, and we recommend”
Line 498: change “we recommend to avoid the use of” to “we recommend avoiding the use of”

---

## Round 0.2 · Minor Revisions

Dear Drs. Rossine and Riina,
The reviewers found your manuscript significantly improved. They only suggest a few small corrections before acceptance. Please make corrections and respond point by point to the reviewers' suggestions to speed up the process.
Thank you
Gabriele

·

Basic reporting

I am generally satisfied that the authors have taken the reviewers' comments under due consideration. There is just one aspect that I believe needs correcting, namely the two citations of iNaturalist records. I see two such records cited for Estado Oaxaca in Mexico. The locality and collection date information is fine, but instead of the collector name (this is sometimes not provided in the iNaturalist profiles, as in the case of "laratamutante"), they should cite instead the iNaturalist observation number. The correct format for the first citation is "https://www.inaturalist.org/observations/167788472" in the case of the C. Dominguez-Rodriguez observation. For the second one, listed as E.S.V. Arellanes, I was not able to attribute this to one of the nine observations of Croton anomalus shown in iNaturalist. That is to say, there were two observations agreeing with the cited observation date of June 25, 2023, but I couldn't tell which of the two they are citing here, since iNaturalist uses pseudonyms, and not all observers provide their real names or other relevant information. The authors should be able to look up the observation in question and then provide the observation number for this second record in the format shown above.

Experimental design

no comment

Validity of the findings

no comment

·

Basic reporting

The authors have addressed my concerns with the original submission. I have a few suggested changes to grammar, punctuation, or wording.

Line 391: Change “conditions” to “condition.”

Line 409: Change “flower of” to “flower with.”

Line 419: Change “revision” to “review.”

Line 445: Change “representative” to “complete.”

Line 462-465: Because a phrase following a semicolon must have both a subject and a verb, change this sentence to “This species is characterized by a set of character states present in
species of section Lasiogyne (van Ee et al., 2011; Rossine et al., 2023), specifically a stellate indumentum...multifid styles.” or insert “these form” after the semicolon. I prefer the former alternative.

Line 468: Delete the comma after “leaves.”

Line 472: Insert a comma before “which.”

Line 494: Change the comma after “internally” to a semicolon.

Line 500: Change the comma after “column” to a semicolon.

Figure 4: There is a discrepancy between the figure caption and the legend regarding the dot colors for previously known records and new records. The figure legend is correct, whereas the caption has them reversed. Furthermore, the legend omits the phrase “and previous records of synonyms,” which is important for the reader to understand; I highly recommend this phrase be added to the legend. Also, the color of the darker dots looks black, not dark blue, on my screen; I recommend changing the color and description to black.

Figure 5: “Croton” should be spelled out in the figure title, not abbreviated. Als, because the two maps are labeled A and B, that information should be included in the figure caption (“Distribution maps of C. blanchetianus (map A, red circles) and C. jacobinensis (map B, black circles)....”).

Table 5: I recommend that the inflorescences of Croton anomalus be described as “Unisexual (staminate) or bisexual” as in the text.

Experimental design

No comment.

Validity of the findings

No comment

Additional comments

Thank you for addressing the reviewers' comments.

---

## Round 0.3 · accepted · Accept

Dear Drs. Rossine and Riina,
All the reviewer's concerns are satisfied. So, I am pleased to inform you that your paper " Untangling the taxonomic knot of Croton anomalus (Euphorbiaceae), a Neotropical dry forest shrub" is accepted for publication in the PeerJ. Congratulations!
Thank you for submitting your work to PeerJ.
Sincerely,
Gabriele Casazza